# A Systematic Review of Clinical Trials on the Efficacy and Safety of CRLX101 Cyclodextrin-Based Nanomedicine for Cancer Treatment

**DOI:** 10.3390/pharmaceutics15071824

**Published:** 2023-06-26

**Authors:** Ana Serrano-Martínez, Desirée Victoria-Montesinos, Ana María García-Muñoz, Pilar Hernández-Sánchez, Carmen Lucas-Abellán, Rebeca González-Louzao

**Affiliations:** Faculty of Pharmacy and Nutrition, Campus de los Jerónimos, Universidad Católica San Antonio de Murcia (UCAM), 30107 Guadalupe, Murcia, Spain; aserrano@ucam.edu (A.S.-M.); phsanchez@ucam.edu (P.H.-S.); clucas@ucam.edu (C.L.-A.); rgonzalez6@ucam.edu (R.G.-L.)

**Keywords:** CRLX101, cancer treatment, targeted therapy, cyclodextrin

## Abstract

CRLX101 is a cyclodextrin-based nanopharmaceutical designed to improve the delivery and efficacy of the anti-cancer drug camptothecin. Cyclodextrins have unique properties that can enhance drug solubility, stability, and bioavailability, making them an attractive option for drug delivery. The use of cyclodextrin-based nanoparticles can potentially reduce toxicity and increase the therapeutic index compared to conventional chemotherapy. CRLX101 has shown promise in preclinical studies, demonstrating enhanced tumor targeting and prolonged drug release. This systematic review followed PRISMA guidelines, assessing the efficacy and toxicity of CRLX101 in cancer treatment using clinical trials. Studies from January 2010 to April 2023 were searched in PubMed, Scopus, Web of Science, and Cochrane Database of Systematic Reviews, using specific search terms. The risk of bias was assessed using ROBINS-I and Cochrane risk-of-bias tools. After screening 6018 articles, 9 were included in the final review. These studies, conducted between 2013 and 2022, focused on patients with advanced or metastatic cancer resistant to standard therapies. CRLX101 was often combined with other therapeutic agents, resulting in improvements such as increased progression-free survival and clinical benefit rates. Toxicity was generally manageable, with common adverse events including fatigue, nausea, and anemia.

## 1. Introduction

Cancer is the second leading cause of morbidity and mortality worldwide, affecting millions of people each year [1]. It is distinguished by its unusual and unchecked cell growth that can occur in any part of the body, presenting in over 100 distinct variations. The most prevalent types include lung, breast, prostate, colorectal, and gastric cancers [2]. The incidence of cancer fluctuates based on geographic location, demographic variables, and lifestyle habits [3]. This illness has a profound influence on public health due to its high occurrence rate, the costs associated with its treatment and healthcare, as well as the emotional strain it places on patients and their loved ones [4]. It also impinges on patients’ quality of life, with many suffering from debilitating symptoms and side effects from treatment, such as fatigue, pain, and impaired ability to carry out daily tasks [5]. Further, a social stigma often accompanies a cancer diagnosis, influencing the working, social, and emotional aspects of the lives of those affected [6].

Conventional cancer treatments include surgery, radiotherapy, and chemotherapy [7]. Surgery is used to remove solid tumors and can be curative if performed in early disease stages [7]. Radiotherapy employs ionizing radiation to destroy cancer cells and reduce tumors, often in combination with other treatments [8]. Chemotherapy uses drugs that target rapidly growing cells, including cancerous ones, but can also affect healthy cells, causing side effects [9]. Conversely, emerging cancer therapies, such as immunotherapy and targeted therapy, have gained ground in recent years [10]. Immunotherapy harnesses the patient’s immune system to fight cancer, using monoclonal antibodies, checkpoint inhibitors, and cellular therapies, such as chimeric antigen receptor T cells (CAR-T) [11]. Targeted therapy, on the other hand, employs drugs that specifically act on molecules or cellular pathways involved in cancer cell growth and survival, minimizing damage to healthy cells [12]. In this landscape of targeted therapy, cyclodextrins (CDs) have emerged as a promising tool in the formulation of anti-cancer drugs, thus making their history and specific properties crucial to understand.

CDs belong to a family of naturally derived cyclic oligosaccharides that are sourced from starch. They consist of six (known as α-), seven (referred to as β-), or eight (termed γ-cyclodextrins) glucose units. These units are interconnected through α(1→4) glycosidic bonds [13]. These molecules have a frustoconical structure with an internal hydrophobic cavity and an external hydrophilic surface, enabling them to form inclusion complexes with a wide variety of substances, enhancing solubility and stability [14]. Cyclodextrins have proven useful in pharmaceutical formulation, especially for improving bioavailability and drug delivery of poorly water-soluble drugs [15]. Motivated by their properties and the urgent need to overcome the limitations of conventional anticancer drugs such as toxicity and multidrug resistance, cyclodextrins have been employed in the development of new cancer treatments [16]. Several types of cyclodextrins, including α-, β-, and γ-cyclodextrin, as well as their hydroxypropyl and methylated derivatives, have been used in cancer drug formulation [17]. These cyclodextrins can form inclusion complexes with anticancer drugs, enhancing solubility, stability, and bioavailability, which may lead to increased efficacy and reduced toxicity [18]. 

CRLX101 serves as a prime example of the potential of cyclodextrin-based nanomedicines in cancer therapy, but it is not alone. Other formulations utilizing the versatility of cyclodextrins have also shown promising results in clinical settings. For instance, NK012, an SN-38-incorporating polymeric micelle, is formed by the self-assembly of a block copolymer composed of polyethylene glycol (PEG) and poly(glutamic acid), using the inherent properties of cyclodextrins [19]. The results from a phase II study indicated that NK012 is effective against advanced, recurrent, or metastatic colorectal cancer and well-tolerated [20].

Thus, the development and clinical investigation of CRLX101, along with formulations such as NK012, underscore the significant potential of cyclodextrin-based nanomedicines in cancer therapy. By enhancing drug delivery and therapeutic efficacy while mitigating toxicity, these groundbreaking treatments herald a promising future for more effective and personalized cancer therapeutic strategies.

CRLX101 is an innovative nanomedicine that was developed using a unique method called “self-assembly”, where a cyclodextrin-containing polymer, camptothecin (CPT), and a linker molecule spontaneously form a nanoparticle drug conjugate [21]. The drug CPT is covalently linked to the cyclodextrin-containing polymer through a cleavable linker, creating a drug-polymer conjugate. This conjugate then self-assembles into nanoparticles, encapsulating additional CPT within the nanoparticle core [22]. At present, CRLX101 is progressing through the final phases of clinical trials. Its potential effectiveness has been explored across a diverse range of cancer types, such as lung, kidney, and ovarian cancer [23]. 

CRLX101 presents a significant edge over conventional CPT formulations due to its capacity to extend the duration of cancer cell exposure to the active pharmaceutical ingredient. Thanks to its nanoparticle construction, CRLX101 enables a gradual, sustained discharge of CPT, thereby augmenting its anti-cancer impacts and elevating the therapeutic index [24]. Moreover, by focusing on the tumor microenvironment rather than specific cancer cells, CRLX101 may circumvent prevalent drug resistance mechanisms, a substantial constraint of many traditional anti-cancer medications [25]. The selective accumulation of CRLX101 within tumors, coupled with the sustained drug release, could help lessen systemic exposure and toxicity often associated with free CPT [26].

Nevertheless, CRLX101 is not without its limitations. Despite the unique formulation enhancing drug delivery and curbing systemic toxicity, patients can still experience side effects such as fatigue, diarrhea, and neutropenia. Additionally, it is not uncommon for some tumors to exhibit resistance to CRLX101, whether inherent or acquired [27]. In instances of acquired resistance, prolonged usage of CRLX101 may induce modifications in tumor cells, reducing their susceptibility to the drug. Similarly, in the case of inherent resistance, some tumors may demonstrate initial resistance to CRLX101 attributable to intertumoral genetic variations [28].

Beyond CRLX101, cyclodextrins have been utilized in devising other drug delivery systems for cancer therapies, encompassing nanoparticles, liposomes, and micelles [17]. Drug delivery systems based on cyclodextrin, in combination with an array of therapeutic agents, including doxorubicin, paclitaxel, and cisplatin, have shown enhanced therapeutic efficacy and diminished side effects [18,29]. Leveraging cyclodextrins in the formulation of anti-cancer drugs and controlled-release systems presents an encouraging strategy to bolster the effectiveness and safety of oncological treatments.

Thus, this systematic review of clinical trials using CRLX101 for cancer treatment aspires to assess and compile the existing evidence concerning the efficacy and safety of CRLX101. The goal is to offer a more in-depth understanding of its therapeutic potential, pinpoint areas for refinement and optimization, and contribute meaningful insights for future investigations and the creation of personalized oncological treatments using cyclodextrins as a therapeutic platform.

## 2. Materials and Methods

This comprehensive review was carried out in adherence to the guidelines set forth by the Preferred Reporting Items for Systematic Review and Meta-Analyses [30]. The scope of the clinical trials included in this research endeavor was to evaluate the potency of CRLX101 in cancer management and gauge its potential toxic effects. Our systematic review and meta-analysis have been duly registered with the International Prospective Register of Systematic Reviews (PROSPERO) under the registration number CRD42023424511.

### 2.1. Eligibility Criteria

The following inclusion criteria were established: (a) participants—cancer patients undergoing treatment; (b) outcomes—measures of potential cancer improvement and toxicity of CRLX101; and (c) clinical trials. Searching was restricted to articles in English or Spanish language published in peer-reviewed journals.

The exclusion criteria were (a) studies comparing CRLX101 with other therapeutic agents or combinations without evaluating the impact on cancer; (b) review articles, case reports, and cross-sectional or longitudinal studies; (c) studies involving patients without cancer or not measuring the effect of therapy on cancer; and (d) duplicated studies.

### 2.2. Information Sources and Search Strategy

The investigators A.M.G.-M. and D.V.-M. executed a comprehensive search across various databases, namely PubMed, Scopus, Web of Science, and the Cochrane Database of Systematic Reviews. This was carried out within a specific timeframe, from January 2010 to April 2023. The decision to include studies in our review was based on factors such as participants involved, outcomes observed, and criteria for inclusion and exclusion. We employed a myriad of search terminologies to identify relevant information. These included terms related to the study’s focus: (a) “cyclodextrins”, “CRLX101”, “NLG207”, “beta-cyclodextrin”, “gamma-cyclodextrin”, “alpha-cyclodextrin”, “hydroxypropyl-beta-cyclodextrin”, “methyl-beta-cyclodextrin”, “sulfobutyl ether-beta-cyclodextrin”; (b) “cancer”, “neoplasms”, “tumor”, “carcinoma”, “malignancy”, “oncology”; (c) “drug delivery”, “drug administration”, “chemotherapy”, “targeted therapy”, “nanoparticles”, “nanomedicine”, and “drug carriers”. To ensure thoroughness, the search terminologies were customized to match each individual database and were run through the specific search parameters provided by these databases.

### 2.3. Selection Process

The identification of appropriate studies was followed by the utilization of Mendeley (Version for Windows 10; Elsevier, Amsterdam, The Netherlands) to remove duplicates. This process was independently executed by two investigators (A.M.G.-M. and D.V.-M.), who reviewed each title and abstract to determine potential articles requiring full-text examination. To address any disagreements, a third researcher, A.S.-M., was involved.

### 2.4. Data Items and Quality Assessment

In this research, the task of extracting specific variables such as patient improvements post CRLX101 intravenous treatment, toxicity levels, average age, type of clinical trial, patient’s health status, trial phase, and dosage was assigned to one researcher (D.V.-M.). A separate researcher (A.M.G.-M.) was responsible for confirming the accuracy of this data. Any disagreements between these two researchers were resolved by bringing in a third researcher (A.S.-M.) for further review.

For assessing the potential risk of bias within the studies, we utilized two different tools: the ROBINS-I tool (Risk of Bias in Non-randomized Studies—of Interventions) [31] for non-randomized studies, and the Cochrane risk-of-bias tool (RoB 2.0) [32] for randomized clinical trials. Each of these tools serves a specific purpose; ROBINS-I examines the presence of bias through seven categories, while RoB 2.0 evaluates bias across five different categories. This bias risk assessment for the included studies was performed independently by two researchers, D.V.-M. and A.M.G.-M.

In more detail, ROBINS-I scrutinizes study bias through seven distinct domains: confounding factors, participant selection, intervention classification, deviations from planned interventions, handling of missing data, the way outcomes are measured, and the reporting of results. On the other hand, the RoB 2.0 tool focuses on five aspects of bias: bias stemming from the process of randomization, deviations from intended interventions, missing outcome data, the measurement methodology of the outcome, and selection in reporting results.

## 3. Results

### 3.1. Search Results

We conducted a systematic review of clinical trials to assess the effectiveness and improvements produced by CRLX101 in cancer patients as well as its toxicity. The search and selection strategy employed in the systematic review followed the Preferred Reporting Items for Systematic Reviews and Meta-Analyses (PRISMA) guidelines and is illustrated in Figure 1. A total of 6018 articles were initially identified from various databases: SCOPUS (*n* = 2927), Web of Science (*n* = 1789), PubMed (*n* = 1282), and Cochrane (*n* = 20). Among these, 2742 duplicates were removed, and the titles and abstracts of the remaining 3276 articles were reviewed. After this step, 20 articles were selected for further assessment. Upon full-text examination, 11 articles were excluded for various reasons: 4 being abstracts [33,34,35,36], 2 involving mice studies [23,24], 3 not measuring the effect on cancer [37,38,39], 1 being a duplicate [33], and 1 being a review [40]. This left nine articles for the final systematic review.

### 3.2. Characteristics and Quality Assessment of Included Articles

The main characteristics of the nine included studies are summarized in Table 1. The articles were published between 2013 and 2022. A total of 323 participants (50.3% women) had a mean age ranging from 59 to 76 years. All studies focused on patients with advanced, metastatic, or inoperable cancer conditions resistant to standard therapies. Various therapies were employed in combination with CRLX101. All studies were conducted in the USA, except for Voss et al. [41], which was carried out in both the USA and Korea. The studies included in this systematic review share several common characteristics, such as investigating the therapeutic potential of CRLX101 in a range of cancer types at various stages, administering the drug in different dosing regimens, and assessing its impact on tumor progression, clinical outcomes, and associated toxicities (Table 1). Most of the studies were conducted in the context of phase I/II clinical trials or pilot studies. Specifically, Gaur et al. [26], Keefe et al. [42], and Weiss et al. [43] were phase I/IIa studies, while Duska et al. [44] and Sanoff et al. [45] were phase Ib/II studies. Krasner et al. [46], Schmidt et al. [47], and Voss et al. [41] were all phase II trials, and Chao et al. [48] conducted a single-center pilot trial.

These studies focused on advanced or metastatic malignancies, encompassing gastric, ovarian, renal cell, and prostate cancers, among others. CRLX101 was administered at different dosages, and several studies shared similar dosing regimens. Chao et al. [48] and Voss et al. [41] used 15 mg/m^2^, Duska et al. [44] tested three dose levels (9, 12, and 15 mg/m^2^), and Gaur et al. [26] tested multiple dosages (6, 12, 15, and 18 mg/m^2^). Keefe et al. [42] utilized escalating doses of 12 and 15 mg/m^2^, while Krasner et al. [46] administered 15 mg/m^2^ every 21 or 28 days. Sanoff et al. [45] used two dosing phases of 15 mg/m^2^, Schmidt et al. [47] administered 12 mg/m^2^, and Weiss et al. [43] administered 15 mg/m^2^ bi-weekly.

On the other hand, in the studies, patients had a variety of conditions. Chao et al. [48] included patients resistant to at least one type of systemic treatment for advanced, inoperable, or metastasized stomach, gastroesophageal junction, or esophageal squamous cell or adenocarcinoma. Gaur et al. [26] studied patients with metastatic or unresectable solid tumor malignancies refractory to standard curative therapy. Duska et al. [44] focused on women with epithelial ovarian cancer (EOC). Keefe et al. [42] investigated patients with metastatic or locally advanced unresectable renal cell carcinoma who received a median of two prior therapies, including at least one prior vascular endothelial tyrosine kinase inhibitor therapy (VEGF-TKI). Krasner et al. [46] studied patients with recurrent epithelial ovarian, tubal, or primary peritoneal cancer who had up to three prior lines of treatment for Cohort A and up to two prior lines for Cohort B, excluding hormones and prior treatment with PARP inhibitors. Sanoff et al. [45] included patients with locally advanced rectal cancer. Schmidt et al. [47] investigated patients with metastatic castration-resistant prostate cancer who had a median of three prior systemic therapies. Voss et al. [41] examined patients with metastatic renal cell carcinoma (RCC) of any histologic subtype who had received two to three prior lines of molecularly targeted therapy, including at least one VEGF-inhibiting regimen, while Weiss et al. [43] studied patients with advanced solid tumor malignancies.

In these investigations, the drug CRLX101 was often combined with other therapeutic agents. For instance, bevacizumab was used in conjunction with CRLX101 by Duska et al. [44], Keefe et al. [42], Krasner et al. [46], and Voss et al. [41]. Furthermore, standard chemoradiotherapy was employed alongside CRLX101 in the study by Sanoff et al. [45]. Additionally, platinum and fluoropyrimidine therapy was used in combination with CRLX101 by Chao et al. [48].

Moreover, the studies included in the analysis showed some common improvements across various clinical outcomes. Many studies reported increased progression-free survival (PFS), such as Duska et al. [44], Keefe et al. [42], Krasner et al. [46], and Weiss et al. [43]. Additionally, several studies observed clinical benefit rates (CBR) and overall response rates (ORR), as seen in Krasner et al. [46] and Sanoff et al. [45]. Some unique improvements were also noted, such as selective absorption of CRLX101 into gastric tumor tissue and reduced activity of potential drug targets such as carbonic anhydrase IX and HIF-1α in Chao et al. [48] and tumor downstaging in Sanoff et al. [45].

Regarding toxicity, most studies reported a manageable profile with common adverse events (AEs) such as fatigue, nausea, and anemia. Grade ≥ 3 AEs were observed in Keefe et al. [42], including non-infectious cystitis, fatigue, anemia, diarrhea, dizziness, and other individual events. In Krasner et al. [46], hypertension and qualitatively increased bladder toxicities were reported with the addition of bevacizumab, but no severe adverse events (SAEs) occurred. However, Schmidt et al. [47] found that CRLX101 was poorly tolerated in patients with metastatic castration-resistant prostate cancer, with intolerable toxicity attributed to non-infective cystitis.

The risk of bias assessment for the non-randomized studies was conducted using the ROBINS-I tool. The findings from the table indicate the predominant risk of bias for the included studies and highlight areas where biases may potentially influence the results. In this analysis, the predominant risk of bias among the studies was moderate. The domain with the highest risk was “Bias due to confounding”, which could be attributed to uncontrolled or residual confounding factors affecting the study outcomes. The domain with the lowest risk was “Bias in the classification of interventions” and “Bias due to deviations from intended interventions”, suggesting that these aspects of the studies were well handled (Figure 2). In addition to the non-randomized studies, the risk of bias for the randomized study by Voss et al. [41] was also assessed using the RoB 2.0 tool. The assessment indicated a minimal risk of bias across all categories: the process of randomization, divergence from planned interventions, missing outcome data, outcome measurement, and choice of the result reported (Figure 3).

## 4. Discussion

The current analysis aimed to examine the efficacy and safety of CRLX101, a novel nanoparticle drug conjugate, in the treatment of various advanced cancers. Evidence from multiple clinical trials, carried out in diverse patient demographics and under different conditions, have underlined the potential of CRLX101 as a possible therapeutic option for several types of cancer. These insights accentuate the need for more comprehensive studies on the application of CRLX101, either in tandem with other treatments or as a sole therapeutic option. In the following discussion, we will delve into several key aspects of the study outcomes, such as disparities in safety and efficacy, dosage and administration, along with possible rationales for the observed variations between different trials.

### 4.1. Efficacy and Safety of CRLX101 across Different Clinical Trials and Conditions

Considering the observed therapeutic improvements and related toxic side effects in various trials, it is evident that the effectiveness of CRLX101 might differ significantly based on the specific type of cancer, the applied treatment plan, and the patient demographic. For instance, a pilot study conducted by Chao et al. [48] in a single center observed that CRLX101 was selectively absorbed by gastric tumor tissue. This study, involving patients with advanced, inoperable, or metastasized stomach, gastroesophageal junction, or esophageal squamous cell or adenocarcinoma, reported a favorable toxicity profile. In contrast, a study by Schmidt et al. [47] found no notable improvements in patients suffering from metastatic castration-resistant prostate cancer, with the treatment showing low tolerability due to excessive toxicity.

To gain a better understanding of the varying degrees of efficacy and safety associated with CRLX101, it would be beneficial to juxtapose these results with other studies that involve cancer treatment and cyclodextrins. For example, a preclinical study by Hrkach et al. [49] explored the efficacy of a PSMA-targeted docetaxel nanoparticle conjugate in cancer models. While not CRLX101 per se, this approach shares similarities with the overall concept of nanoparticle-based chemotherapy delivery. The study found that the docetaxel nanoparticle conjugate exhibited a differentiated pharmacological profile and enhanced antitumor efficacy compared to free docetaxel in preclinical models. This highlights the potential benefits of combining nanomedicine-based therapeutics with traditional chemotherapy agents in the treatment of advanced cancers. On the other hand, new data from a large-scale genomic study on cyclodextrin-based therapies revealed significant patient-specific variability, hinting at the need for personalized dosing strategies [50].

One possible explanation for the differences in efficacy and safety may be the intrinsic tumor characteristics, such as genetic and molecular profiles, that could influence the response to treatment. For example, in the study by Weiss et al. [43], the median progression-free survival for patients treated at the maximum tolerated dose was 3.7 months, with stable disease observed in 28 patients (64%) and confirmed stable disease in 15 patients (34%). This suggests that the response to CRLX101 may depend on the molecular characteristics of the tumor, which could also impact the overall treatment outcomes. On the other hand, a phase I clinical trial conducted by Mita et al. [51] investigated the safety, tolerability, and pharmacokinetics of cyclodextrin-based nanoparticle-encapsulated paclitaxel (IT-101) in patients with advanced solid tumors. The study reported promising safety and preliminary antitumor activity, although the responses varied across different cancer types and patient populations. A study by Anselmo et al. [52] further showed how nanocarriers can alter the immunological response, indicating a complex interplay between the nanodrug and the patient’s immune system.

Considering these factors, a study by Liu et al. [53] highlighted the importance of understanding the tumor microenvironment and its potential impact on the efficacy of nanomedicine-based therapeutics. The authors suggested that factors such as tumor heterogeneity, stromal components, and immune cell infiltration could significantly affect the delivery and efficacy of nanoparticle-based therapies. This further emphasizes the need for more research to elucidate the complex relationship between tumor characteristics and the effectiveness of CRLX101. Furthermore, Jain et al. [54] emphasized how the tumor microenvironment could pose physical barriers to drug delivery, further complicating the scenario.

Another potential explanation for the differences in efficacy and safety could be the presence of other comorbidities or concurrent therapies in the patient population. For example, in the study by Sanoff et al. [45], patients with locally advanced rectal cancer received standard chemoradiotherapy and radiotherapy, along with concurrent administration of capecitabine, which could have influenced the overall response to CRLX101. Furthermore, the study by Gaur et al. [26] found that the lower toxicity observed in their patient cohort was not due to allele frequency differences in drug metabolism genes, suggesting that other factors, such as concurrent treatments or comorbidities, could be responsible for the observed differences in toxicity. For example, a comprehensive study by the Cancer Genome Atlas Research Network [55] highlighted the influence of patient’s genotypic variations and concurrent medications on the overall effectiveness and safety profile of oncological therapies.

In conclusion, the relatively small number of studies and diversity of patient populations currently available in the literature makes it a challenge to conclusively assess the efficacy of CRLX101 in different conditions.

### 4.2. Dosing and Administration of CRLX101

In optimizing the dose and administration of CRLX101, it is important to consider various aspects to maximize its therapeutic potential and minimize adverse effects. The varying degrees of efficacy and safety observed across different trials and conditions may be attributed to differences in dosing and administration as well as the intrinsic characteristics of the tumor and the patients receiving the treatment.

A general trend observed in CRLX101 clinical trials is that lower doses administered more frequently, such as 12 mg/m² every 2 weeks, tend to result in a more favorable toxicity profile [48]. Conversely, higher doses administered less frequently, such as 15 mg/m² every 3 weeks, have been associated with a higher incidence of intolerable toxicity [47]. This observation is consistent with the findings of Weiss et al. [43], who determined the maximum tolerated dose (MTD) of CRLX101 to be 12 mg/m² when administered every 3 weeks in their phase II study involving patients with advanced solid tumors. The efficacy of CRLX101 may also be influenced by its combination with other anticancer agents. In a study by Gaur et al. [26], the combination of CRLX101 with bevacizumab demonstrated promising response rates and a manageable safety profile in patients with metastatic renal cell carcinoma. 

When comparing CRLX101 with the broader literature on nanoparticle-based drug delivery, it becomes apparent that the optimal dosing regimen for CRLX101 may depend on several factors, including the cancer type, treatment regimen, and patient population. For example, studies on other nanoparticle-based therapeutics, such as those utilizing liposomes or polymeric micelles, have also shown that lower doses administered more frequently tend to result in improved efficacy and reduced toxicity [56,57]. This finding is consistent across various cancer types and may be attributed to the unique properties of nanoparticle-based drug delivery systems.

One such property is the enhanced permeability and retention (EPR) effect, which allows nanoparticles to preferentially accumulate in tumor tissue due to the leaky vasculature and impaired lymphatic drainage often observed in tumors [58]. This effect may influence the optimal dosing regimen for CRLX101 and other cyclodextrin-based therapies, as higher drug concentrations in the tumor tissue could lead to increased efficacy and reduced systemic toxicity [59]. This observation is echoed in a review by Maeda et al. [60], where they discuss the impact of the EPR effect on nanoparticle drug delivery and how it might be leveraged to improve treatment outcomes.

Moreover, it is increasingly recognized that specific tumor characteristics, such as molecular and genetic profiles, could influence the response to treatment and, consequently, the optimal dosing regimen [61]. For instance, a study by Boehnke et al. [62] highlighted the role of tumor genetics in determining the efficacy of nanoparticle-based drugs. They found that tumors with certain genetic profiles were more likely to respond to nanoparticle-based therapies, suggesting that a personalized approach to treatment could improve outcomes for patients. As such, it becomes imperative to consider individual patient factors and tumor characteristics when determining the most appropriate dosing strategy for CRLX101.

### 4.3. Limitations and Future Directions

While the findings from the various clinical trials of CRLX101 offer valuable insights into the potential benefits of this cyclodextrin-based therapy for cancer treatment, several limitations must be acknowledged. These limitations should be considered when interpreting the results and planning future research.

First, the number of clinical trials and patient populations studied for CRLX101 is relatively small. This limits the generalizability of the results and makes it difficult to draw definitive conclusions about the effectiveness and safety of CRLX101 across different cancer types and patient populations. More extensive clinical trials with larger and more diverse patient populations will be required to further validate the efficacy and safety of CRLX101. Second, the studies included in the analysis had varying designs, methodologies, and endpoints, which may have contributed to the observed heterogeneity in the results. Future studies should aim for a more standardized approach in terms of trial design and outcome assessment to enable more accurate comparisons between studies and a more comprehensive understanding of the treatment effects. Third, many of the trials were early-phase studies, which primarily focused on safety, tolerability, and pharmacokinetics rather than clinical efficacy. To establish the therapeutic benefits of CRLX101, more advanced trials (e.g., phase II or III) with larger patient cohorts and well-defined clinical endpoints are necessary. Lastly, the underlying mechanisms through which CRLX101 exerts its antitumor effects are not yet fully understood. A deeper understanding of the molecular and cellular mechanisms that drive the efficacy and safety of CRLX101 could help optimize the treatment strategy and potentially improve patient outcomes. Further preclinical and clinical research is needed to elucidate these mechanisms and identify potential biomarkers that could predict treatment response and aid in patient selection.

## 5. Conclusions

In this systematic review, we analyzed the potential of CRLX101, a cyclodextrin-based nanopharmaceutical, as a therapeutic agent for various cancer types. Our findings suggest that the unique properties of cyclodextrin-based nanoparticles might provide certain advantages over conventional chemotherapy, possibly enhancing the therapeutic index and reducing systemic toxicity. However, the efficacy and safety of CRLX101 appear to be influenced by several factors, including cancer type, treatment regimen, patient population, and dosing strategy. While these preliminary findings are encouraging, further research is necessary to fully understand the clinical implications of CRLX101 in the treatment of cancer.

## Figures and Tables

**Figure 1 pharmaceutics-15-01824-f001:**
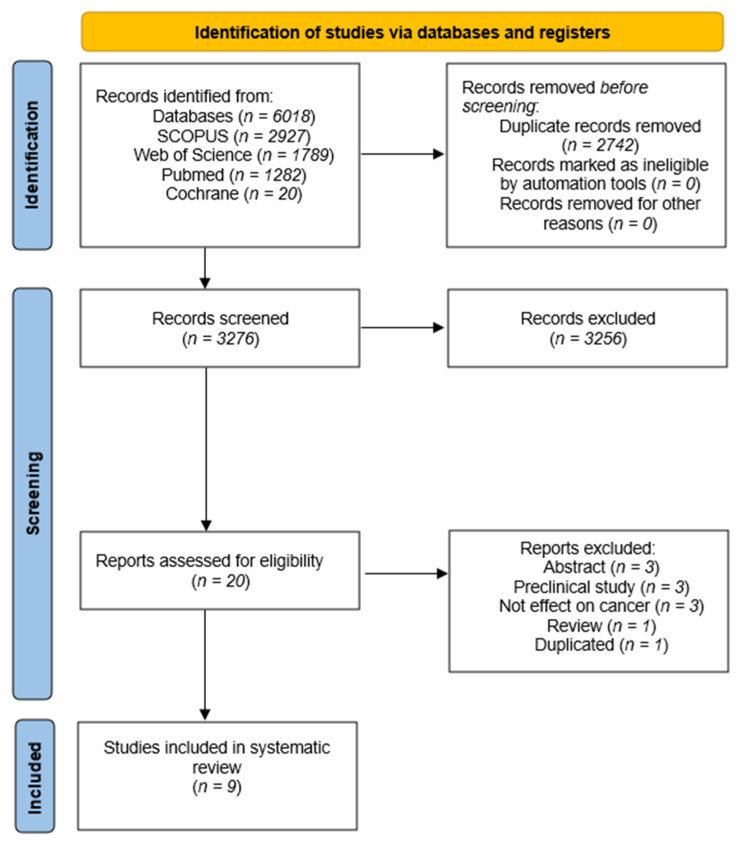
Flow diagram of the systematic review.

**Figure 2 pharmaceutics-15-01824-f002:**
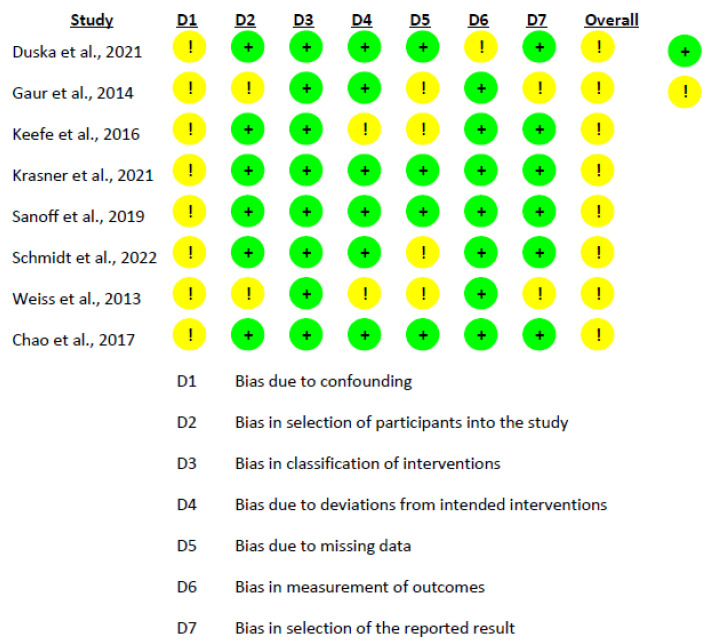
Risk of Bias Assessment using ROBINS-I for the Included Intervention Studies [26,42,43,44,45,46,47,48].

**Figure 3 pharmaceutics-15-01824-f003:**
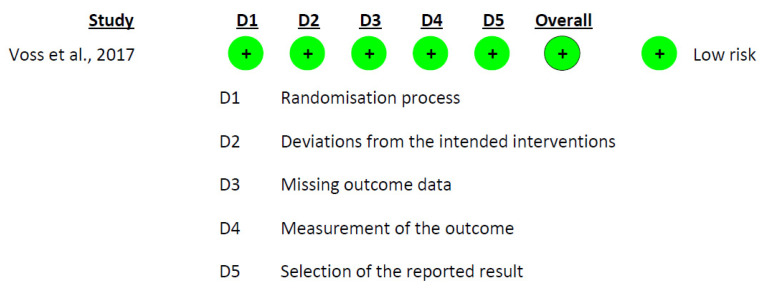
Risk-of-Bias Assessment Using RoB 2.0 for the Included Intervention Studies [41].

**Table 1 pharmaceutics-15-01824-t001:** Characteristics of the studies included (*n* = 9).

Author	Year	Clinical Trial Phase	*n*	Women (%)	Mean Age	Condition	Therapy	Dose	Improvement	Toxicity
Chao et al. [48]	2017	Pilot	10	60	64	Advanced stomach, gastroesophageal junction, or esophageal squamous cell or adenocarcinoma	CRLX101 + Platinum and fluoropyrimidine therapy	CRLX101 dosed at 15 mg/m^2^ on days 1 and 15 of a 28-day cycle	Signs of selective absorption and reduced activity of potential drug targets	Favorable toxicity profile
Gaur et al. [26]	2014	1/2a	NR	NR	NR	Metastatic or unresectable solid-tumor malignancies	CRLX101 + Standard therapy	CRLX101 dosed at 6, 12, 15 or 18 mg/m^2^ infused over a 60-min period on days 1, 8, and 15 of each 28-day cycle	Inhibition of topoisomerase expression and associated with longer survival duration	Lower toxicity not due to allele frequency difference in drug metabolism genes
Duska et al. [44]	2021	1b/2	19	100	62	EOC in women	CRLX101 + Bevacizumab treatment	Three CRLX101 dose levels: starting (12 mg/m^2^), one escalation (15 mg/m^2^) and one de-escalation (9 mg/m^2^). The weekly paclitaxel dose was fixed (80 mg/m^2^). The RP2D of EP0057 was established	Similar efficacy in platinum sensitive and resistant patients	The combination of EP0057 and weekly paclitaxel was found to be tolerable
Keefe et al. [42]	2016	1/2a	22	18	63	Metastatic or locally advanced unresectable renal cell carcinoma	CRLX101 + at least one prior vascular endothelial tyrosine kinase inhibitor therapy	Escalating doses of CRLX101 (12, 15 mg/m^2^) in a 3 + 3 phase I design. An expansion cohort of 10 patients was treated at the RP2D. Patients with refractory mRCC were treated every 2 weeks with bevacizumab (10 mg/kg) and CRLX101.	23% partial responses, 55% achieved PFS > 4 months	Grade ≥ 3 AEs related to CRLX101
Krasner et al. [46]	2021	2	63	100	61	Recurrent epithelial ovarian, tubal or primary peritoneal cancer.	CRLX101 + up to 3 prior lines of treatment	CRLX101 was administered at a dose of 15 mg/m^2^ every 21 days for Cohort A and 15 mg/m^2^ every 28 days for Cohort B, with the addition of bevacizumab at a dose of 10 mg/kg every 14 days for Cohort B.	CBR of 68–95%, PFS of 4.5–6.5 months	Well tolerated, increased hypertension and bladder toxicities with bevacizumab
Sanoff et al. [45]	2019	1b/2	32	NR	59	Locally advanced rectal cancer	CRLX101 + standard chemoradiotherapy	Two dosing phases of CRLX101 (every other week and weekly) of 15 mg/m^2^	50% downstaging of the primary tumor, 68% nodal site downstaging	Most common AE were fatigue and lymphopenia
Schmidt et al. [47]	2022	2	4	0	76	Metastatic castration-resistant prostate cancer	CRLX101 + median of 3 (range 3–4) prior systemic therapies	CRLX101 was administered at a dose of 12 mg/m^2^ every 2 weeks	No improvements	Poorly tolerated and not feasible due to intolerable toxicity (noninfective cystitis) attributed to CRLX101 in patients with mCRPC
Voss et al. [41]	2017	2	110	24	NR	mRCC of any histologic subtype	CRLX101 + 2–3 prior lines of molecularly targeted therapy	CRLX101 15 mg/m^2^ intravenous on days 1 and 15 of a 28-day cycle	No added benefit with combination therapy	Toxicity observed in the phase II study was manageable and consistent with the phase I experience. The most common TEAEs for CRLX101 plus bevacizumab included fatigue, nausea, constipation, decreased appetite, and headache
Weiss et al. [43]	2013	1/2a	62	50	63	Advanced solid-tumor malignancies	CRLX101 + chemotherapy, radiotherapy, or other investigational therapy	CRLX101 15 mg/m^2^ administered bi-weekly by intravenous infusion	Median PFS: 3.7 months, 64% SD	Fatigue, cystitis, anemia, neutropenia, nausea, dysuria, hematuria, leukopenia

AEs: adverse events; CBR: clinical benefit rate; EOC: epithelial ovarian cancer; mCRPC: metastatic castration-resistant prostate cancer; mRCC: metastatic renal cell carcinoma; NR: not reported; PFS: progression-free survival; RP2D: recommended phase 2 dose; SD: stable disease; TEAEs: treatment-emergent adverse events.

## Data Availability

Not applicable.

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
