# Peer review of "A Systematic Review of Clinical Trials on the Efficacy and Safety of CRLX101 Cyclodextrin-Based Nanomedicine for Cancer Treatment"

_pharmaceutics, 2023, doi:10.3390/pharmaceutics15071824_

Round 1

Reviewer 1 Report

The manuscript entitled A Systematic Review of Clinical Trials on the Efficacy and Safety of CRLX101 Cyclodextrin-Based Nanomedicine for Cancer Treatment by D. Victoria-Montesinos, A. M. García-Muñoz and coworkers, contains an elaborate literature review analyzing potential of CRLX101, a cyclodextrin-based nanopharmaceutical, as a therapeutic agent for various cancer types. The work is well structured, the choice of articles has been described by the authors and the subject reported in the text is clear. In my view, the paper can be published without further modifications.

Just a little correction: table 1, column “Dose”, mg/m2, superscript "2"

Reviewer 2 Report

This review focused on the clinical trials on the efficacy and safety of CRLX101. The content of this review is appropriate and meaningful. However, this manuscript needs a major revision before its publication. My comments are as follows:

1.       Further background about CRLX101 should be introduced, such as its preparation process, development phase, advantages/ disadvantages to other camptothecin formulations.

2.       More details about the advantages of Cyclodextrin-Based Nanomedicine (not limited to CRLX101) in clinical are suggested in a separate section.

3.       Only 9 articles were included in the final review, the number of clinical trials and patient populations is too small to reveal the efficacy and safety of CRLX101. Besides the articles, further data from the websites, databases and other ways are suggested to be discussed.

4.       Please improve the quality of the figures, especially “Figure 2”. 

Minor editing of English language required

Reviewer 3 Report

Manuscript reviews the application of CRLX101 to cancer treatment. this is review article, but its is hard to read it as a review article. followings are comments,

1) please reconstruct the structure of article. introduction, history, motivation of CRLX101 to cancer, specific properties, actual application, like this.

2) in view of chemistry, structure information, specific interaction of CRLX101 to cancer are poorly written.

3) some articles were surveyed to understand the properties. they are a way to summarize the data.

4) Table had difficulty to understand. please mention more clearly and shortly.

Round 2

Reviewer 2 Report

The authors revised the manuscript carefully according to the comments of the reviewers, I suggest accept this version.

Reviewer 3 Report

Manuscript is revised against the reviewer's comments.